# A Comparative Study on the Debittering of Kinnow (*Citrus reticulate* L.) Peels: Microbial, Chemical, and Ultrasound-Assisted Microbial Treatment

**Shweta Suri** [1] **, Anupama Singh** [1,*] **, Prabhat K. Nema** [1] **and Neetu Kumra Taneja** [2]

1   Department of Food Engineering, National Institute of Food Technology Entrepreneurship and Management, Sonipat 131028, Haryana, India
2   Department of Basic and Applied Sciences, National Institute of Food Technology Entrepreneurship and Management, Sonipat 131028, Haryana, India
*   Correspondence: asingh3niftem@gmail.com

**Abstract:** Kinnow mandarin (*Citrus reticulate* L.) peels are a storehouse of well-known bioactive compounds, viz., polyphenols, flavonoids, carotenoids, limonoids, and tocopherol, which exhibit an effective antioxidant capacity. However, naringin is the most predominant bitter flavanone compound found in Kinnow peels that causes their bitterness. It prohibits the effective utilization of peels in food-based products. In the present study, a novel approach for the debittering of Kinnow peels has been established to tackle this problem. A comparative evaluation of the different debittering methods (chemical, microbial, and ultrasound-assisted microbial treatments) used on Kinnow peel naringin and bioactive compounds was conducted. Among the chemical and microbial method; solid-state fermentation with A. niger led to greater extraction of naringin content (7.08 mg/g) from kinnow peels. Moreover, the numerical process optimization of ultrasound-assisted microbial debittering was performed by the Box–Behnken design (BBD) of a response surface methodology to maximize naringin hydrolysis. Among all three debittering methods, ultrasound-assisted microbial debittering led to a greater hydrolysis of naringin content and reduced processing time. The optimum conditions were ultrasound temperature (40 °C), time (30 min), and *A. niger* koji extract (1.45%) for the maximum extraction rate of naringin (11.91 mg/g). These debittered Kinnow peels can be utilized as raw material to develop therapeutic food products having a high phytochemical composition without any off-flavors or bitterness.

**Keywords:** Kinnow peel; debittering; ultrasound; *Aspergillus niger*; naringin; optimization





## 1. Introduction

Citrus fruits belong to one of the prime, cultivated fruit families in the world, with an annual, global, harvested area of 10.07 million hectares and a production rate of 158.49 million MT [1]. The main citrus fruits cultivated by Indian farmers are sweet lime (*Mosambi*), mandarin (*Kinnow*), oranges, tangerines, bitter orange (*Narangi*), lemon, lime, and grapefruits, which belong to the *Rutaceae* family [2]. The main peculiarities of citrus fruits are their delicious taste, flavor, nutritional content, and phytochemical diversity [3]. Owing to its exotic taste, the Kinnow mandarin (*Citrus reticulate* L.), which is a hybrid variety of King (*Citrus nobilis*) and willow leaf (*Citrus delicosa*), is highly cultivated in the northern states of India with an annual cultivation rate of 10.48 million tonnes [4]. A total of about 75% of Kinnow mandarin is usually consumed in its fresh form and the rest undergoes processing in the food industry for the production of juice, resulting in the generation of high quantities of processing waste in the form of peels, pulp, seeds, and pomace [5]. Kinnow peels are major by-products of Kinnow processing, accounting for 50–55% of the total fruit mass [6]. The inappropriate dumping of Kinnow processing waste can cause many ecological issues, owing to its high moisture level and sugar content [7].

Nonetheless, Kinnow peels comprise a group of bioactive components, such as polyphenols, flavonoids, carotenoids, limonoids, and tocopherol, which exhibit effective antioxidant properties [8]. The flavonoids found in Kinnow peels, such as naringin, hesperidin, nobiletin, limonin, narirutin, tangeretin, and eriocitrin, possess several health advantages that include antioxidant, anti-inflammatory, anti-cancer, anti-diabetic, anti-ulcer, and anti-mutagenic effects [9], while naringin (4′,5,7-thrihydroxyflavanone-7-rhamnoglucoside) is the key flavonoid found in Kinnow peels with high biological activity. However, the naringin flavanone is also considered as one of the most dominant bitter compounds in Kinnow peels that primarily causes their bitterness [2], thereby affecting the utilization and supplementation of the bioactive compounds of Kinnow peels in the food industry. Bioconversion by utilizing enzyme/bacteria and physical adsorption through resin methods are employed to reduce bitterness [10]. Some investigations conducted in the past concentrated on the debittering of citrus fruit juice by enzymatic and exchange resins [11]; microporous resins [12,13]; and amberlite IRA resin [14]. Naringin can be hydrolyzed by the naringinase enzyme into rhamnose and prunin, which can undergo further hydrolysis to form the bitterless compounds of naringenin and glucose by the β-D-glucosidase enzyme. Yet, the research has shown that *Aspergillus niger*, generally recognized as a safe (GRAS) fungi, can also hydrolyze naringin into non-bitter compounds [13]. Additionally, some reports have emphasized the chemical methods used for the effective utilization of citrus-fruit-processing waste [15].

At present, ultrasound technology is gaining importance and can be integrated with other methods to increase the efficiency of food processes. It is widely recognized as a green, energy-saving method, due to its ease of use, low installation and maintenance costs, and minimal energy requirements [16]. It is not only used as a method for enhancing mass transfer, but also in aiding the debittering process. To date, comprehensive reports have proposed that the chemical, mechanical, as well as cavitation effects of ultrasound waves can help modify the structure of the enzyme and its substrate, thereby enabling the interaction of the enzyme and its substrate and also reducing their activation energy, ultimately improving the enzymatic hydrolysis of the substrate and the rate of the enzymatic reaction [17,18]. Moreover, Sharma et al. [19] employed ultrasound treatment at varying temperatures and times (72 °C/16 s; 100 °C/10 min; 100 °C/15 min) to debitter the juice of Giloy (*Tinospora cardifolia*), and observed that the bitterness of Giloy juice was reduced, along with presenting an improvement to the antioxidant activity. Likewise, Gao et al. [20] conducted the debittering of Ougan (*Citrus suavissima Hort. ex Tanaka*) juice through which sonication, together with microbial (*A. niger* koji) treatment (40 kHz, 80 W/L and 60 min), showed an improvement to the hydrolysis efficiency of the enzyme and a decrease in the hydrolysis time of bitterness-causing compounds in Ougan juice.

To date, no study has been conducted on the debittering of Kinnow mandarin peels and the effects of different debittering methods (chemical, microbial, and ultrasound-assisted) on Kinnow mandarin peel bioactive compounds for their effective utilization in the food industry. Furthermore, the research on the influence of chemical, microbial, and ultrasound-assisted treatments on the polyphenols (total phenol and flavonoid) and total antioxidant activity is scarce. Keeping this in view, the present study aims to provide an effective process for the debittering of Kinnow mandarin peels and to verify the effect of different methods on the debittering of Kinnow peels. This will help in establishing a roadmap for the better utilization of Kinnow processing waste in response to Sustainable Development Goal-12 of the United Nations: sustainable production and consumption. Furthermore, the optimization of process parameters of the ultrasound-assisted microbial debittering of Kinnow mandarin peels using designed experiments (Box–Behnken design of RSM) is also performed.

## 2. Materials and Methods

### 2.1. Raw Materials and Chemical Compounds

Fresh Kinnow Mandarin (*Citrus reticulate* L.) peels were obtained from the local juice vendors of Kundli, Sonipat, India. *Aspergillus niger*-224 was procured from the National Collection of Dairy Cultures (NCDC), NDRI, India. Naringin (HPLC grade), methanol (HPLC grade), aluminum chloride, sodium carbonate, gallic acid, Folin–Ciocalteu reagent, ferric chloride, sodium hydroxide, sodium nitrate, and DPPH (2,2-diphenyl-1-picrylhydrazyl) were bought from Sigma Aldrich, India and were of AR grade. Yeast and mold agar 51089 (YM 016) was purchased from the HiMedia laboratory, Mumbai, India.

### 2.2. Preparation of Samples

Kinnow peels (flavedo and albedo) were cleaned, sorted, and washed with potable water to remove the impurities, dust, dirt, pesticide residues, etc. The peels were then crushed into a pulverized form in a laboratory processor (Inalsa Food Processor Model: Inox 1000 Plus) for further operations. The fresh Kinnow peel sample was analyzed for naringin and bioactive compounds using the standard procedures.

### 2.3. Debittering of Kinnow Peels

The new approach toward the debittering of Kinnow peels was established by comparing the chemical, microbial, and ultrasound-assisted microbial debittering processes. In addition, the influence of different treatments on the bioactive profile (total phenol content (TPC), total flavonoid content (TFC), as well as total antioxidant activity (TAA)) of the Kinnow peels was determined. The detailed methodology of the debittering experiments is explained in the following subsections.

#### 2.3.1. Chemical Debittering of Kinnow Peels

Chemical debittering of the Kinnow peels was conducted using the method proposed by Wang et al. [21] following slight modifications. Debittering was performed through the sequential process of alkali treatment, followed by acidic treatment.. The debittering operation was performed by washing the pulverized peels in 5% alkali solution (NaOH solution) for varying time intervals (30, 60, 90, and 120 min) at 40 °C in a water bath (Sanco 220/230 V Waterbath). In the alkali treatment, pulverized Kinnow peels were mixed with NaOH at a ratio of 1:40 (*w/v*). Subsequently, the alkali-treated peels were neutralized with 1% acid solution (citric acid solution) for a period of 120 min at 40 °C in a water bath. The peel sample was then washed with distilled water prior to chemical analysis.

#### 2.3.2. Microbial Debittering of Kinnow Peels by Aspergillus Niger Koji

Microbial debittering was conducted using *Aspergillus niger* culture, which was grown in yeast and mold agar 51089 (YM 016). The media preparation was conducted as per the manufacturer's recommendations by autoclaving at 121 °C, with 15 psi/103,421 pascal pressure for 15 min under static conditions. The debittering of pulverized Kinnow peels was performed by inoculating the *A. niger* spores as explained by Gao et al. [20] with slight changes. Solid-state fermentation (SSF) and submerged fermentation (SMF) processes with *A. niger* were performed. In the solid-state fermentation process, wheat bran (20 g), wheat flour (5 g), pulverized Kinnow peels (3 g), and deionized water (20 mL) were mixed and autoclaved at 121 °C for 20 min. After cooling the mixture to 40 °C, three loops of the *A. niger* spores were inoculated. The prepared mixture was incubated at 30 °C for a period of 96 h. The prepared *A. niger* koji (10 g) was extracted with citrate buffer (20 mL) in a shaker bath at 40 °C for 30 min. Lastly, the resultant extract was filtered through Whatman No.1 filter paper prior to chemical analysis. In the submerged fermentation, the amount of deionized water used to prepare the media was 100 mL, but other than that, the culture process of *A. Niger* koji was similar to solid-state fermentation.

2.3.3. Ultrasound-Assisted Microbial Debittering of Kinnow Peels

Experimental Design for Ultrasound-Assisted Microbial Debittering

An ultrasonicator water bath (Branson 3800 Ultrasonic Bath) (200 W, 40 kHz) was used for performing the ultrasound-assisted microbial debittering of Kinnow peels. Kinnow peels: distilled water (1:10) was obtained for the debittering treatments [7]. A known aliquot of the sample (50 mL) was mixed with different fractions of *A. Niger* Koji (1–3%) with subsequent sonication in a 100 mL beaker at a certain temperature (40–60 °C) and time (30–90 min). Following sonication, the extract obtained was centrifuged at 5000 rpm for 15 min, followed by filtration with Whatman No.1 filter paper. The extract obtained after filtration was kept at −18 °C for further chemical analysis. The fresh Kinnow peel extract without *A. Niger* Koji treatment was used as the control sample.

Ultrasound treatment was conducted to check the effect of sonication on the microbial debittered sample. The optimization of ultrasound-assisted microbial debittering parameters was performed through the Box–Behnken design (BBD) of response surface methodology. A total of 17 experiments were planned with 3 independent parameters, viz, ultrasound-treatment time, temperature, and *A. niger* koji extract at 3 levels, as presented in Table 1. The levels of independent parameters were wisely selected after conducting the preliminary trials. The degree of naringin hydrolysis, total phenol content (TPC), total flavonoid content (TFC), as well as total antioxidant activity (TAA) were used as the dependent variables for the sonication experiments. The overall experimental design for ultrasound-assisted microbial debittering with coded as well as actual values is shown in the Supplementary Table S1.

**Table 1.** Levels of independent parameters for sonication experiment.

| S. No. | Independent Variables | Code | Values of Independent Variables | | |
| --- | --- | --- | --- | --- | --- |
| | | | −1 | 0 | +1 |
| 1. | Temperature (°C) | $X_1$ | 40 | 50 | 60 |
| 2. | Time (Min) | $X_2$ | 30 | 60 | 90 |
| 3. | *Aspergillus Niger* Koji extract (%) | $X_3$ | 1% | 2% | 3% |

*2.4. Chemical Analysis*

2.4.1. Quantitative Investigation of Naringin by HPLC

The naringin content of the debittered Kinnow peel was determined by the method presented by Seal [22] with some changes. The sample extract of the Kinnow peel was passed through a 0.2 μm PTFE filter and thereafter injected into HPLC (HPLC, Agilent Infinity-II, Santa Clara, CA, USA) with a C18 column. The mobile phase for naringin analysis consisted of 0.1% of orthophosphoric acid in water (*v/v*), which was presented as a solvent A and acetonitrile (which was presented as solvent B) having a column temperature of 30 °C. Moreover, the wavelength of the detector was set between 200 and 400 nm. A flow rate of 1 mL/min was induced for analysis, while the wavelength of the UV detector was fixed at 254 nm. The presence of naringin was confirmed by comparing the retention time and area of the sample with the naringin standard, and the naringin in the Kinnow peel samples was quantified accordingly.

2.4.2. Total Phenol Content (TPC)

The Folin–Ciocalteu reagent (FCR) method was employed to determine the total phenols observed in the sample extract of fresh and debittered Kinnow peels [23]. In brief, the debittered sample extract (20 μL) was obtained in a test tube and the volume was made up to 1000 μL with distilled water. Furthermore, 1000 μL of FCR (1:10) and 800 μL of sodium bicarbonate (10%) were added. Thereafter, the mixture was vortexed and incubated in the dark for 30 min for proper color development. The intensity of the developed color in the sample extract and standard (gallic acid) was read at 765 nm using a UV-VIS

spectrophotometer (UV-1800, Shimadzu, Japan). TPC in Kinnow peels was reported as mg gallic acid equivalent (GAE)/g dry weight.

### 2.4.3. Total Flavonoid Content (TFC)

The fresh and debittered Kinnow peel extract was analyzed for TFC using a standard procedure. A sample extract (40 μL) was collected in a test tube and the volume was made up to 1000 μL with distilled water; thereafter, 300 μL of sodium nitrate (5%) was added. After 5 min of incubation, 600 μL of aluminum chloride (10%) and 2000 μL of sodium hydroxide (1N) were added to the above mixture. Furthermore, the sample mixture was vortexed and absorbance was read at 510 nm. Quercetin was presented as the standard for TFC analysis. The flavonoid content of the Kinnow peels was stated as mg quercetin equivalent (QE)/g dry weight [24].

### 2.4.4. Total Antioxidant Activity (TAA)

The TAA of the sample extract was examined based on the percent inhibition of the 2, 2-diphenyl-1-picrylhydrazyl (DPPH) radical [25]. A known volume of sample extract (100 μL) was mixed with methanol (900 μL) to produce a volume of 1000 μL. DPPH solution (3000 μL) was added to the sample extract and mixed thoroughly prior to the incubation at room temperature for 30 min in a dark area. A control sample containing the methanol and DPPH solution was also prepared. The absorbance of the control and sample extract was read at 517 nm. Percent inhibition was determined according to Equation (1).

$$Percent\ Inhibition = \frac{A_C - A_S}{A_C} \times 100 \tag{1}$$

where $A_c$ stands for the absorbance of the control (methanol as control), and $A_s$ stands for the absorbance of the sample extract.

### 2.5. Optimization

The Design-Expert Software package (Version 10.0.8) was utilized to examine the influence of process variables on the degree of naringin hydrolysis and bioactive attributes by employing the one-way analysis of variance (ANOVA) procedure. The second-order polynomial equation was formed to describe the effect of independent factors on the dependent ones (Equation (2)).

$$Y = \beta_0 \pm \beta_1 X_1 \pm \beta_2 X_2 \pm \beta_3 X_3 \pm \beta_{12} X_1 X_2 \pm \beta_{13} X_1 X_3 \pm \beta_{23} X_2 X_3 \pm \beta_{22} X_{22} \tag{2}$$

where $Y$ denotes the study responses (naringin content, total phenol content (TPC), total flavonoid content (TFC), as well as total antioxidant activity (TAA)); $\beta_0$, $\beta_1$, $\beta_2$, $\beta_3$, $\beta_{12}$, $\beta_{13}$, $\beta_{23}$, and $\beta_{22}$ denote the coefficients for intercept, linear, interactive, and quadratic effects; and $X_1$, $X_2$, and $X_3$ denote the independent parameters ($X_1$ = temperature, $X_2$ = time, $X_3$ = *Aspergillus niger* koji extract).

### 2.6. Model Validation

The appropriateness of the established models elucidating the influence of *A. Niger* Koji (1–3%), temperature (40–60 °C), and time (30–90 min.) on the responses (Naringin content, TPC, TFC, and TAA of the Kinnow peels) were confirmed with the optimum settings predicted by design expert software. The error percentages were determined to estimate the "fit of the model" (Equation (3) [26]:

$$Error\ (\%) = \frac{1}{n_e} \sum_{i=1}^{n} \left\| \frac{V_E - V_P}{V_E} \right\| \times 100 \tag{3}$$

where $V_E$ denotes the experimental/actual value; $V_P$ denotes the predicted/expected value obtained after optimization.

## 3. Results and Discussion

### 3.1. Chemical Composition of Kinnow Mandarin Peels before Debittering Treatments

The bioactive composition and naringin content of the Kinnow mandarin peels before debittering treatments were analyzed. Initially, the fresh Kinnow peels contained a naringin content of 19.23 mg/g; TPC, TFC, and TAA of 31.05 mg GAE/g; 25.91 mg QE/g; and 56.57 ± 0.82%, respectively. The bioactive composition of the Kinnow peels was in accordance with the research previously conducted by Rafiq et al. [27] who mentioned a TPC of 24.51 mg GAE/g and Deng et al. [28] who observed a TFC of 25.07 mg RE/g.

### 3.2. Debittering of Kinnow Peels

Naringin, a flavonoid, is the main bitterness-causing compound in Kinnow peels that decreases the sensory acceptability of food products containing Kinnow peels. Therefore, the debittering of Kinnow peels is of great importance for enhancing their acceptability in the food sector. Different debittering methods, viz, chemical treatment with alkali followed by acid solutions; microbial treatment with *A. niger* in solid-state and submerged processes; and microbial-assisted ultrasound treatment, were utilized to check their efficiency of hydrolyzing naringin, thereby reducing the bitterness of the Kinnow peels.

#### 3.2.1. Debittering by Chemical Method

The effect of the administration of 5% alkali with subsequent treatment with food-grade acid (1% citric acid) on the degree of hydrolysis of naringin in Kinnow peels is displayed in Figure 1. The chemical treatment of Kinnow peels resulted in naringin content (6.57 ± 0.33 mg/g), TPC (21.16 mg GAE/g), TFC (10.59 mg QE/g), and TAA (34.36%). In comparison to the fresh Kinnow peels containing 19.23 mg/g naringin content, the chemically debittered Kinnow peels exhibited 6.57 mg/g naringin content. In addition, chemical debittering led to the lowering of the TPC, TFC, and TAA of the debittered peels. In agreement with the results, previous studies indicate that chemical treatment has a considerable influence on bioactive compounds, such as flavonoids and total phenols [29].

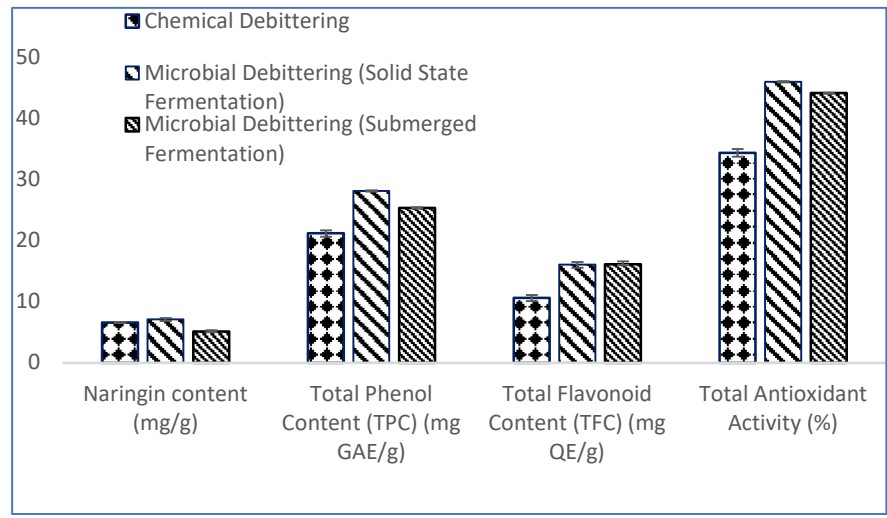

**Figure 1.** Naringin and bioactive components obtained from chemical and microbial methods.

#### 3.2.2. Debittering by Microbial Method (by Treatment with *A. niger* koji)

The effect of *A. niger* koji on the naringin content of Kinnow peels is shown in Figure 1. Two methods of fermentation, viz, solid-state fermentation (SSF), as well as submerged fermentation (SMF) with *A. niger* koji, were performed to check the efficiency of hydrolyzing naringin in Kinnow peels. The study reported that submerged state fermentation (SMF) of Kinnow peels with *A. niger* koji showed a naringin content of 5.09 mg/g, TPC of 28.11 mg GAE/g, TFC of 16.02 mg QE/g, and TAA of 45.99%, while solid-state fermentation (SSF) reported the naringin content, TPC, TFC, and TAA of 7.08 mg/g, 25.33 mg GAE/g,

16.09 mg QE/g, and 44.15%, respectively. The results show that SSF with *A. niger* koji led to a greater extraction of naringin from Kinnow peels as compared to SMF with *A. niger* koji. Furthermore, in comparison to the fresh Kinnow peels, microbially debittered Kinnow peels showed a lower naringin content. In SSF, fermentation occurs in the complete absence or a small amount of free water, thereby resembling the natural environment to which microorganisms adapt. In addition, SSF results in the better effectual biosynthesis of the enzymes under settings without catabolite repression [30]. Furthermore, high levels of TPC, TFC, and TAA were also observed in the SSF of Kinnow peels with *A. niger* koji, indicating the efficacy of microbial treatment.

3.2.3. Debittering Performed by an Ultrasound-Assisted Microbial Debittering Process

Ultrasound is a new-age, green, energy-saving technology that helps in reducing processing steps and confirming food safety. It also helps in the release of bioactive compounds from the cellular matrix into the solvent by the cavitation bubble collapse effect. By ultrasonically extracting flavonoids, viz, naringin from the citrus peels, these flavonoid compounds can be released in a simple and reproducible manner [31]; therefore, it is important to study the effect of microbiological methods combined with ultrasound on the debittering of Kinnow peels.

Model Characteristics and Validation

The influence of independent parameters on the responses is presented in Table 2. According to ANOVA, the F value for response surface models is significant ($p < 0.05$ and $< 0.01$) and the CV value is lower than 10%, which shows that the models fit adequately for every process parameter (Table 3). It can be observed that all the models have $R^2$ (coefficient of determination) values greater than 0.91, which demonstrates that all the analyzed polynomial models correlate well with the measured figures. Moreover, the predicted $R^2$ value is the same as the adjusted $R^2$, with a difference of below 0.2 between the two indicating the adequacy of the model. Adequate precision for all the models was observed to be high, which indicated that the models were appropriate to direct the design space.

**Table 2.** Experimental values for naringin content, total phenolic content (TPC), total flavonoid content (TFC), and total antioxidant activity (TAA) of Kinnow peel extracts obtained through varying debittering parameters.

| Run | Temperature (°C) | Time (min) | *Aspergillus niger* Koji Extract (%) | Naringin Content (mg/g) | TPC (mg GAE/g) | TFC (mg QE/g) | TAA (%) |
|---|---|---|---|---|---|---|---|
| 1 | 50 | 30 | 3 | 8.91 | 17.12 | 12.44 | 30.79 |
| 2 | 40 | 60 | 3 | 16.60 | 15.32 | 15.39 | 30.83 |
| 3 | 50 | 90 | 1 | 13.02 | 16.80 | 19.28 | 31.09 |
| 4 | 60 | 60 | 1 | 18.23 | 21.89 | 16.42 | 37.83 |
| 5 | 50 | 30 | 1 | 14.10 | 30.30 | 26.66 | 45.32 |
| 6 | 40 | 60 | 1 | 15.45 | 31.57 | 26.12 | 38.81 |
| 7 | 50 | 60 | 2 | 13.02 | 22.02 | 16.89 | 43.06 |
| 8 | 50 | 60 | 2 | 14.53 | 20.32 | 16.82 | 44.06 |
| 9 | 40 | 90 | 2 | 5.19 | 21.31 | 22.18 | 45.71 |
| 10 | 50 | 60 | 2 | 14.91 | 20.53 | 15.92 | 47.31 |
| 11 | 40 | 30 | 2 | 12.33 | 25.94 | 26.42 | 48.59 |
| 12 | 50 | 60 | 2 | 14.80 | 21.40 | 16.02 | 54.73 |
| 13 | 50 | 60 | 2 | 14.30 | 20.32 | 16.40 | 54.98 |
| 14 | 60 | 30 | 2 | 5.32 | 24.22 | 19.86 | 57.07 |
| 15 | 50 | 90 | 3 | 7.35 | 14.87 | 13.50 | 58.43 |
| 16 | 60 | 60 | 3 | 8.61 | 16.52 | 11.88 | 58.96 |
| 17 | 60 | 90 | 2 | 5.02 | 15.03 | 14.95 | 61.70 |

TPC: total phenol content; TFC: total flavonoid content; TAA: total antioxidant activity.

**Table 3.** Analysis of variance (ANOVA) table for the fit of independent factors to the response surface model.

| Factors | Product Responses | | | |
|---|---|---|---|---|
| | Naringin Content (mg/g) | TPC (mg GAE/g) | TFC (mg QE/g) | TAA (%) |
| Model F-value | 34.33 *** | 36.92 *** | 58.76 * | 7.94 ** |
| Lack of fit | 2.35 ns | 3.33 ns | 6.51 ns | 0.18 ns |
| CV (%) | 8.05 | 5.16 | 4.51 | 9.93 |
| $R^2$ | 0.9778 | 0.9794 | 0.9869 | 0.9108 |
| Adjusted $R^2$ | 0.9494 | 0.9528 | 0.9701 | 0.7960 |
| Predicted $R^2$ | 0.7613 | 0.7550 | 0.8230 | 0.7058 |
| Model adequate precision | 18.1695 | 20.0454 | 24.9342 | 9.5839 |

*** Significant at $p < 0.001$; ** significant at $p < 0.01$; * significant at $p < 0.05$; ns—non-significant at $p > 0.05$. TPC: total phenol content; TFC: total flavonoid content; TAA: total antioxidant activity.

### 3.3. Response Analysis

3.3.1. The Influence of Process Parameters on Naringin Content of Debittered Kinnow Peels

Naringin is a well-known bioflavonoid with a chemical formula of 4′, 5, 7-trihydroxy flavonone 7-rhamnoglucoside, and citrus fruits are known to be the richest sources of flavonoids, especially naringin. It is also the main, bitter, flavanone found in citrus peels [32]; hence, the degree of hydrolysis of naringin is considered as an important dependent parameter in microbial debittering associated with ultrasound. The obtained values of naringin content in Kinnow peels varied from 5.02 to 18.23 mg/g; the maximum naringin extraction was achieved when the kinnow peels were treated with 1% *A. niger* koji extract at a temperature of 60 °C for 60 min (Table 2). According to the regression model, the linear and quadratic effects of ultrasound treatment time, temperature, and *A. niger* koji extract, and the interactive effects of temperature and time, as well as temperature and *A. niger* koji extract, were significant ($p < 0.01$; $p < 0.001$) (Table 4). The interactive action of different independent parameters on the naringin content of Kinnow peels was determined through 3D surface plots. As indicated in the response surface models, the naringin content in Kinnow peels was substantially affected by the ultrasound treatment time and temperature (Figure 2a). Naringin content decreased with an increase in the temperature from 40 to 60 °C at a constant treatment time. The lowest level of naringin content (5.35 mg/g) was observed at 60 °C. Previous studies reported that ultrasonication synergistically increased the catalytic activity of the naringinase enzyme that caused the hydrolysis of naringin and reduced the level of bitterness in grapefruit juice [33]. On a similar note, enhanced naringinase activity in *Citrus maxima* fruits at a temperature range of 45–65 °C was reported [34].

Furthermore, from the interaction between the time and *A. niger* koji extract (Figure 2b), it was observed that at a 1% administration of *A. niger* koji extract, as the treatment time increased from 30 to 60 min, there was initially an upsurge in naringin from 14.26 to 18.54 mg/g of extract until 60 min, while, later, a decline in the naringin to 12.00 mg/g of extract was observed with an increase in treatment time. However, at 3% *A. niger* koji extract, when the treatment time increased from 30 to 60 min, a slight increase in naringin from 9.8 to 13.94 mg/g, then a gradual decrease in naringin content to 7.17 mg/g extract was detected. It is well recognized in the older literature that the degree of hydrolysis of naringin increases with the increasing treatment time [35]. The interactive effects of *A. niger* koji extract and temperature on the naringin content of Kinnow peels (Figure 2c) are consistent with the results obtained by Gao et al. [20], who observed the enhanced hydrolysis of naringin in Ougan juice with the usage of 2% *A. niger* extract. In addition, *A. niger* generates high concentrations of naringinase enzymes (a complex containing β-D-glucosidase and α-L-rhamnosidase) through solid state fermentation [36], which leads to the conversion of naringin into naringenin (debittered compound).

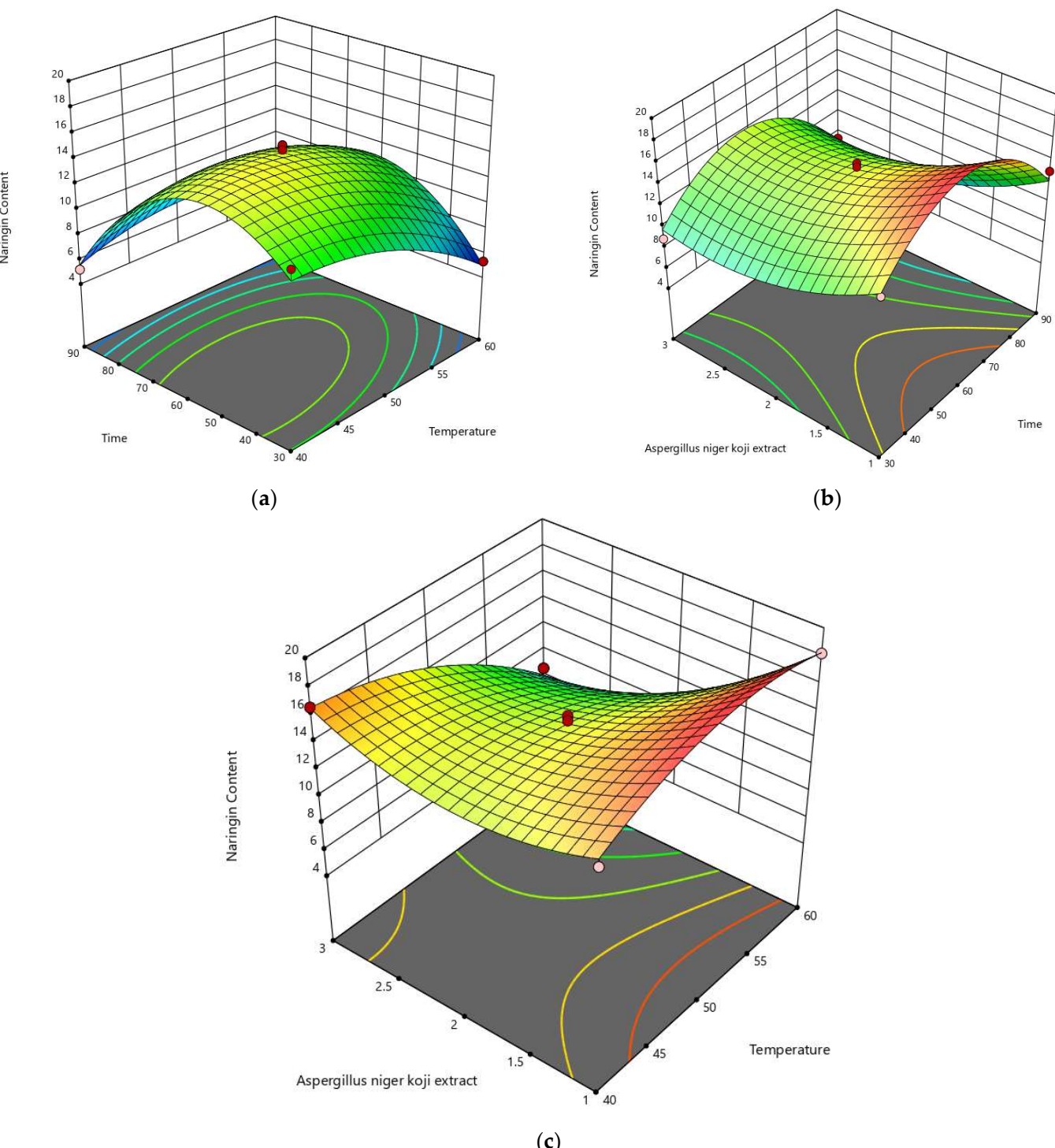

**Figure 2.** Three-dimensional response surface plots for naringin content as affected by processing parameters (**a**–**c**).

**Table 4.** Coefficients of regression analysis for independent process parameters and product responses.

| Effect | Process Parameters Fitted Model-Quadratic | Product Responses | | | |
|---|---|---|---|---|---|
| | | Naringin Content (mg/g) | TPC (mg GAE/g) | TFC (mg QE/g) | TAA (%) |
| Linear | Intercept | 14.31 | 20.92 | 16.41 | 48.83 |
| | $X_1$ Temperature (°C) | −1.55 | −2.06 | −3.37 | 6.45 |
| | $X_2$ Time (Min) | −1.26 | −3.70 | −1.93 | 1.89 |
| | $X_3$ *Aspergillus niger* Koji Extract (%) | −2.42 | −4.59 | −4.41 | 3.25 |
| Interactive | $X_1X_2$ | 1.71 | −1.14 | −0.170 | 1.88 |
| | $X_1X_3$ | −2.69 | 2.72 | +1.55 | 7.27 |
| | $X_2X_3$ | −0.120 | 2.81 | +2.11 | 10.47 |
| Quadratic | $X_1^2$ | −1.73 | 1.13 | +1.96 | 2.32 |
| | $X_2^2$ | −5.61 | −0.42 | +2.48 | 2.12 |
| | $X_3^2$ | 2.15 | −0.72 | −0.923 | −9.54 |
| Fit statistics | Standard deviation | 0.95 | 0.87 | 0.81 | 0.61 |
| | Mean | 11.86 | 20.91 | 18.07 | 46.43 |

TPC: total phenol content; TFC: total flavonoid content; TAA: total antioxidant activity.

3.3.2. The Influence of Process Parameters on the Total Phenol Content (TPC) of Debittered Kinnow Peels

Polyphenols are bioactive compounds that are abundant in fruits, vegetables, and their by-products. These polyphenols (phenols and flavonoids) contribute towards several health benefits for the host [6]; thus, TPC was selected as a preferential dependent parameter for optimization purposes. In the present study, the obtained values for the TPC in Kinnow peels treated with ultrasound-assisted microbial treatment ranged from 14.86–31.57 mg GAE/g, where the highest TPC was reported when the Kinnow peels were treated with 1% *A. niger* koji extract at an ultrasound treatment time of 60 min and 40 °C temperature (Table 2). Based on the regression analysis conducted through Design Expert Software, all three independent parameters, viz, temperature, treatment time, and *A. niger* koji extract, presented a negative linear effect on the TPC of the Kinnow peels ($p < 0.01$; $p < 0.001$). Moreover, a positive ($p < 0.01$) interactive effect of temperature and *A. niger* koji extract, as well as time and *A. niger* koji extract, was observed (Table 4).

The treatment time and temperature presented a negative impact on the TPC of the peels (Figure 3a). Generally, exposure to higher reaction temperatures (60 °C) for longer durations accelerates the oxidative degradation of phenolic compounds. These results are consistent with those obtained by the former research conducted on the ultrasound-assisted extraction of total phenols from rugosa rose (*Rosa rugosa* Thumb.) fruit [37], while the interactive effect of the temperature and *A. niger* koji extract exhibited a significant ($p < 0.01$) positive effect on the TPC (Figure 3b). At 3% *A. koji* extract administration, when the temperature was increased from 40 to 60 °C, the TPC value increased from 16.07 to 17.47 mg GAE/g, which may have been because of the release of polyphenolic complexes from the cellular matrix during microbial fermentation with the increasing temperature. Furthermore, at the interactive level (Figure 3c), the treatment time and *A. niger* koji extract presented a significant ($p < 0.01$) positive influence on the TPC of the Kinnow peels. The increase in the TPC content with the administration of *A. niger* koji extract and increasing treatment time was due to the fact that the conjugated polyphenolic compounds contained in Kinnow peels were released by enzymatic hydrolysis via carbohydrate-metabolizing enzymes (β- glucosidase) produced by *A. niger* during the fermentation of lignocellulosic waste, such as citrus peels [38,39]. In line with the present study, an increase in the TPC through fermentation by fungus was reported for apple waste [39] and mango seed waste [40].

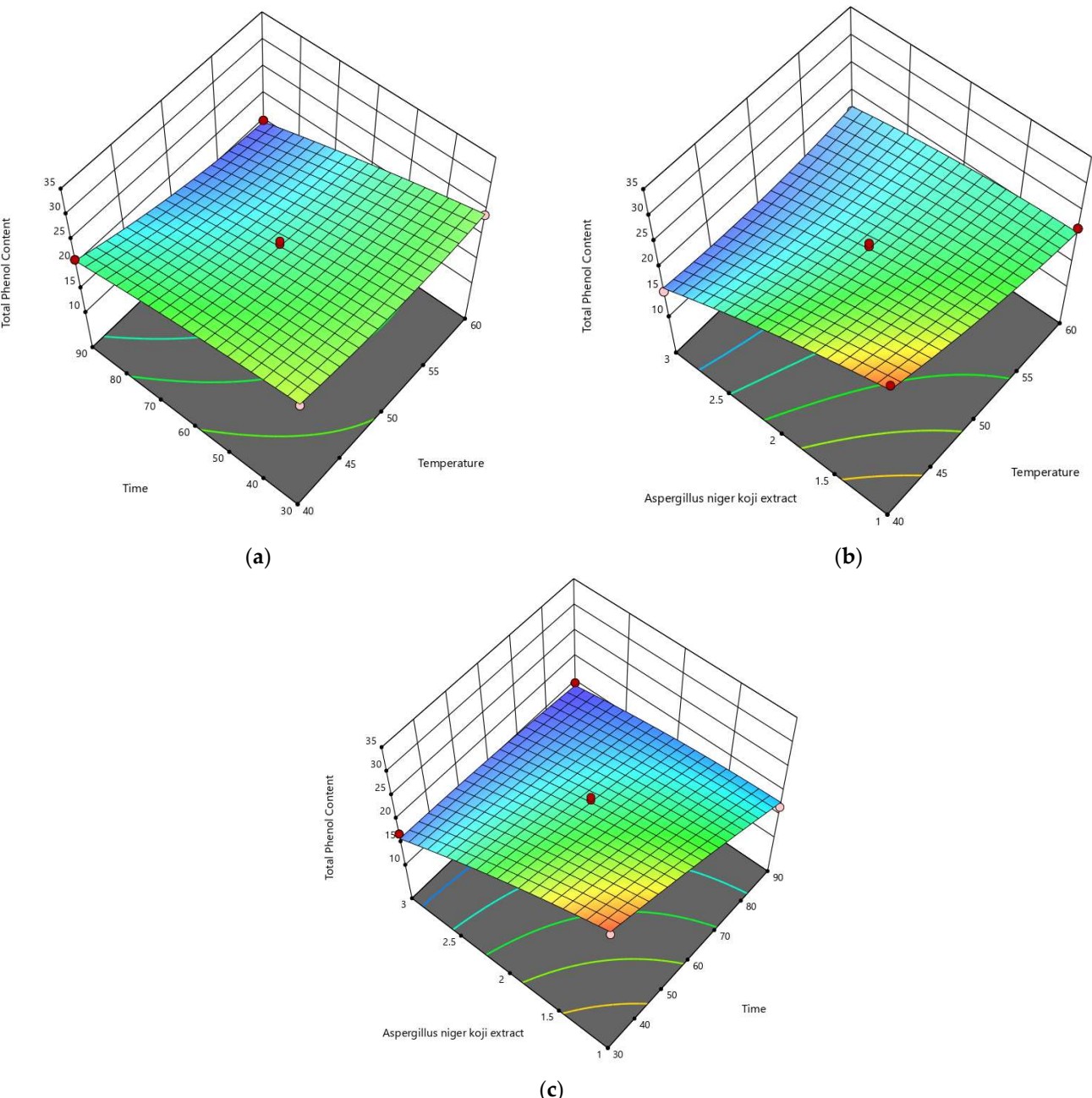

**Figure 3.** Three-dimensional response surface plots for TPC as affected by processing parameters (**a–c**).

### 3.3.3. The Influence of Process Parameters on the Total Flavonoid Content (TFC) of Debittered Kinnow Peels

Flavonoids are polyphenolic complexes found in plant cells and have many chemical and biological properties with a therapeutic potential. In citrus fruit processing, these are widely distributed in pulp, peel, and rag cells [41]. Therefore, it is important to study the changes in the TFC with ultrasonic and microbial treatments. The obtained values for the TFC ranged from 11.87 to 26.66 mg QE/g (Table 2). *A. niger* koji extract readily decreased the TFC of the Kinnow peels (Figure 4a), which was in agreement with the work conducted by Gao et al. [20] on the debittering of Ougan juice by ultrasound coupled with enzymatic hydrolysis. It can be observed that, at a constant treatment time when the concentration of *A. niger* koji extract was increased from 1 to 3%, a decrease in the TFC was also observed. Usually, the flavonoid glycoside's degradation is dependent on the collaborative activities of enzymes, viz, α-L-rhamnosidases and β-glucosidases, produced by *A. niger* [42]. There

are various forms of scientific evidence demonstrating the beneficial and adverse effects of ultrasonic action on the retention/enhancement of bioactive constituents in plant-based materials, though the specificity is dependent on the processing parameters and type of materials [43].

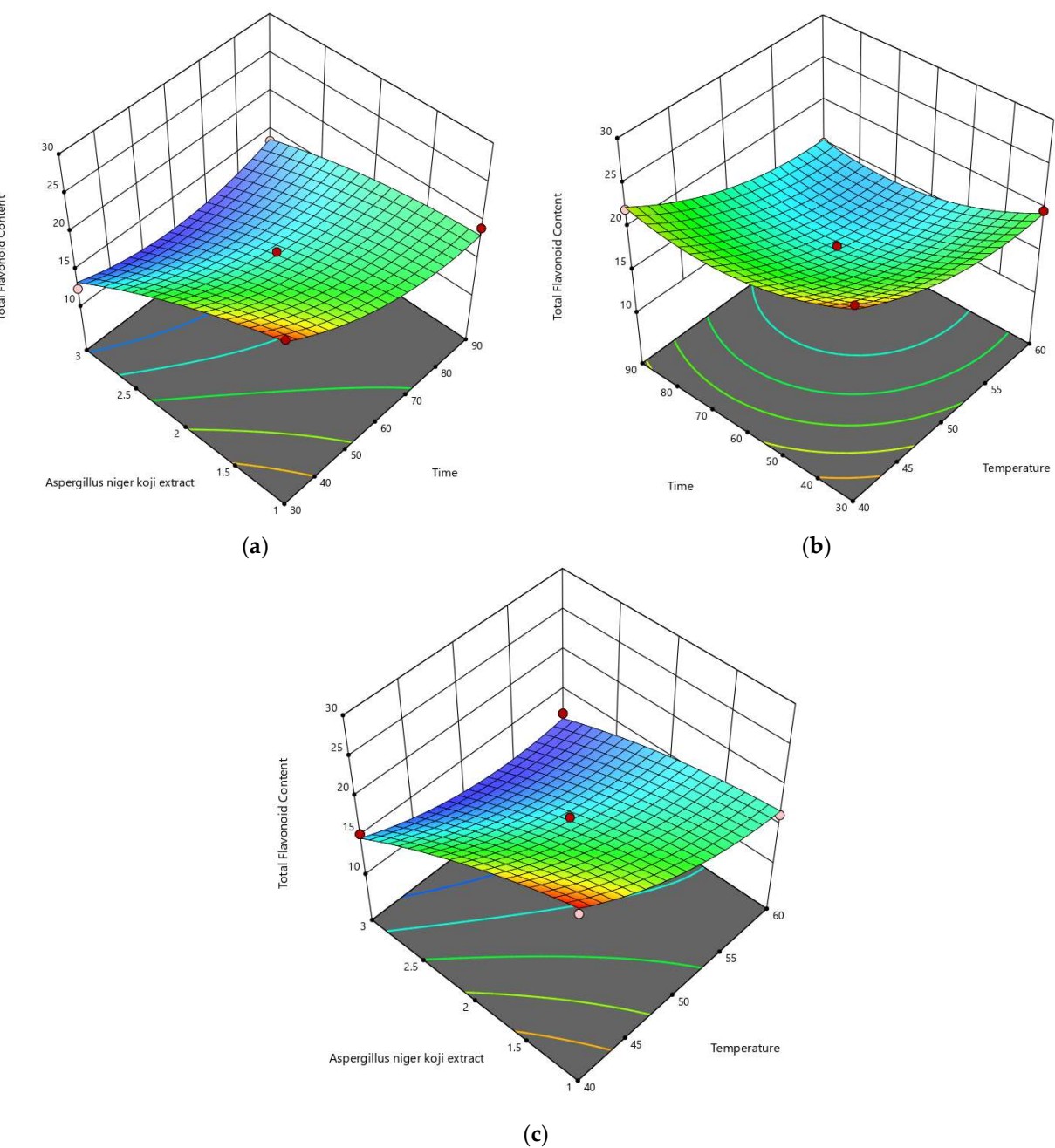

**Figure 4.** Three-dimensional response surface plots for TFC as affected by processing parameters (**a**–**c**).

From the interaction of ultrasound treatment time as well as temperature (Figure 4b), it was observed that, at a constant temperature of 40 °C, as the time increased from 30 to 90 min, a decrease in the TFC (25.49 to 22.33 mg QE/g) was also observed. However, at a higher temperature (60 °C), when the treatment time was increased from 30 to 90 min, a greater decrease in the TFC was observed. The prolonged ultrasound exposure time at higher temperatures led to a reduction in the TFC. In agreement with the present study, Ren et al. [44] reported that a longer exposure time to ultrasound was one of the contributing factors to the leaching or degradation of flavonoids. However, deprivation drift throughout the ultrasound treatment could be associated with the creation of free

radicals, leading to a probable increase in oxidative pathways [45]. Similarly, from the interactive effect of the temperature and *A. niger* koji extract, it was observed that with an increase in the temperature for constant *A. niger* koji extract, the TFC of the Kinnow peels decreased (Figure 4c), where the loss of flavonoids could be a result of flavonoid breakdown, degradation, or decay.

### 3.3.4. The Influence of Process Parameters on the Total Antioxidant Activity (TAA) of Debittered Kinnow Peels

Antioxidant activity refers to the power of bioactive constituents of food to maintain the cellular structure along with the function by efficiently scavenging free radicals, constraining lipid peroxidation, and averting subsequent oxidative damages. Furthermore, the antioxidant activity of foods is the basis of different additional biological functions, for example, anti-cancer, anti-inflammatory, and anti-aging [46]; hence, a thorough analysis of the antioxidant activity of foods is of great importance for the health of human beings. The experimental values for the TAA of treated Kinnow peels ranged from 30.79 to 61.69% (Table 2). Based on the regression analysis, the temperature presented a linear influence on the TAA of Kinnow peels, while a positive interactive effect of time and *A. niger* koji extract as well as temperature and *A. niger* koji extract was observed. At the quadratic level, *A. niger* koji extract exhibited a significant negative effect on TAA (Table 4). As illustrated in Figure 5a, an increase in the TAA of Kinnow peels was observed with a rise in temperature from 40 to 60 °C at a constant treatment time. Primarily, at 40 °C temperature, a 20% increase in TAA, while at 60 °C, a 34.42% enhancement was observed. The findings are in agreement with the research conducted by Kaur et al. [47], where enhancement in the TAA of ultrasound-extracted Kinnow peels was reported with an increase in the temperature. Similarly, another study reported the greater antioxidant activity of tomatoes with increased temperature, which could be associated with the increased extractability of antioxidant constituents following heat treatment [48]. However, another explanation is that at higher temperatures, hydroxyl radicals (OH°) are generated and, by the addition of hydroxyl radicals (OH°) to the ortho or para positions, can enhance the antioxidant activity of bioactive molecules [49], thereby enhancing the overall TAA. From the interaction between *A. niger* koji extract and temperature (Figure 5b), it was noticed that, initially, at a fixed temperature of 40 °C, as the concentration of *A. niger* koji extract increased from 1 to 3%, a slight decrease in the TAA (39.40 to 32.04 %) was observed, whereas the TAA increased to 58.51%, as the temperature of 60 °C was attained. The findings of the present work are in agreement with the former study [40], where a rise in the antioxidant activity of Mexican mango seeds was detected. Furthermore, the interactive effect of treatment time and *A. niger* koji extract (Figure 5c) showed that with an increase in time, at a constant concentration of *A. niger* koji extract, the TAA content first decreased, then increased.

### 3.4. Optimization and Validation of Process Parameters

The process optimization of the ultrasound-assisted microbial debittering of Kinnow peels was conducted to obtain the optimum conditions for independent variables that would help in lowering the naringin content and at the same time maintain the TPC, TFC, and TAA of Kinnow peels. Throughout the process optimization, the response similar to naringin was kept at a minimum level, while TPC, TFC, and TAA were kept at a maximum level. According to the mathematical regression analysis, the solution with the highest desirability (75.60%) was selected for the ultrasound-assisted microbial debittering operation. The optimized values of naringin content (11.91 mg/g), TPC (29.57 mg GAE/g), TFC (30.14 mg GAE/g), and TAA (51.87%) were detected via temperature (40 °C), treatment time (30 min), and *A. niger* koji extract (1.45%). Table 5 indicates the predicted as well as experimental data of the process parameters showing a non-significant difference and low-error percentage among the means of values expressing the effect of treatments in an effective manner. The analyzed data prove that the ultrasound-assisted microbial debittering of Kinnow peels is an efficient way to reduce naringin content while maintaining

bioactive compounds in the Kinnow mandarin peels. In addition, it is also useful in lowering the processing time of debittering experiments to 30 min.

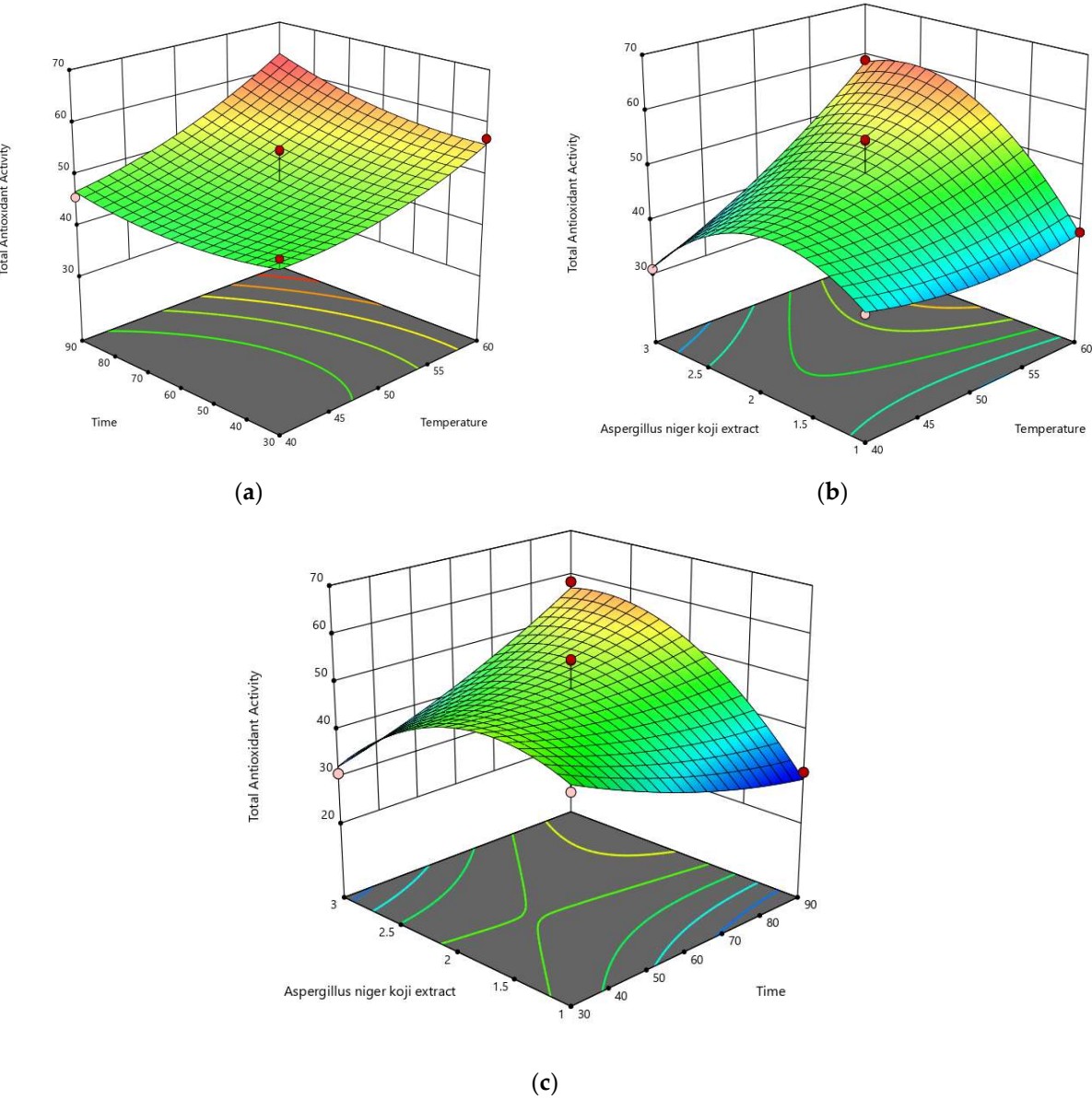

**Figure 5.** Three-dimensional response surface plots for TAA as affected by processing parameters (**a–c**).

**Table 5.** Predicted and experimental values of dependent parameters.

| Particulars | Goal | Optimum Value | Desirability of Model | Error (%) |
|---|---|---|---|---|
| Temperature | Is in range | 40 °C | | |
| Time | Is in range | 30 min | 0.756 | |
| *Aspergillus niger* Koji extract (%) | Is in range | 1.45% | | |
| | | Predicted Value | Experimental Value | |
| Naringin content (mg/g) | Minimize | 14.31 | 14.06 | 1.69 |
| #TPC (mg GAE/g) | Maximize | 20.91 | 21.38 | 2.25 |
| $TFC (mg QE/g) | Maximize | 16.40 | 16.90 | 3.05 |
| *TAA (%) | Maximize | 48.82 | 49.23 | 0.84 |

#TPC: total phenol content; $TFC: total flavonoid content; *TAA: total antioxidant activity.

It has been observed that the ultrasound-assisted microbial treatment led to higher extraction levels of naringin (11.91 mg/g) due to the enhanced naringinase activity at

optimized conditions. Additionally, compared to the fresh Kinnow peels, the TPC, TFC, and TAA were reduced to 29.57 mg GAE/g, 24.14 mg QE/g, and 51.87%, respectively, which reflected a pattern similar to that mentioned in the previous study [7], where a reduction in the TPC (1.58 mg/g) of Kinnow residues was observed. The lowering of the bioactive composition of Kinnow residue was due to the conversion of naringin (bitter compound) to naringenin (non-bitter compound).

## 4. Conclusions

The current research focused on studying different approaches and proposing a new approach for the debittering of Kinnow mandarin peels for their effective valorization in the food processing industry. To observe the influence of pretreatments on the naringin content, different methods involving chemical (with alkali followed by acid), microbial (with *A. niger* koji), and ultrasound-assisted microbial treatments with *A. niger* koji extract were compared using designed experiments. In conclusion, microbial debittering by solid-state fermentation (SSF) resulted in the greater extraction of naringin (7.08 mg/g) from Kinnow peels. Moreover, the chemical debittering performed by utilizing alkali and acid treatments sequentially resulted in naringin content of 6.57 mg/g.

Ultrasound-assisted microbial debittering with *A. niger* koji extract (1.45%) at optimized settings, viz, ultrasound-treatment time (30 min) and temperature (40 °C), markedly improved the naringin hydrolysis of Kinnow peels along with maintaining the bioactive properties, viz, TPC, TFC, and TAA. The optimum values of naringin content (11.91 mg/g), TPC (29.57 mg GAE/g), TFC (24.14 mg QE/g), and TAA (51.87%) were obtained by optimization using the Box–Behnken design of RSM. Thus, it can be said that out of all the debittering treatments performed in the present work, ultrasound-assisted microbial debittering was found to be more appropriate in hydrolyzing naringin from Kinnow mandarin peels. The debittered Kinnow mandarin peels can be employed as a raw ingredient for the preparation of value-added food products and confectionary items, e.g., bread, biscuits, muffins, etc, as Kinnow mandarin peels contain a good storehouse of bioactive antioxidants, dietary fiber, micronutrients, and pectin that can be used to prepare therapeutic foods without any off-taste or bitterness. This approach will not only add value to the food, but will also pave the way for the effective utilization of peels produced from citrus-fruit-processing industries.

**Supplementary Materials:** The following supporting information can be downloaded at: https://www.mdpi.com/article/10.3390/fermentation8080389/s1, Table S1: Box-behnken design for ultrasound-assisted microbial debittering of kinnow peels.

**Author Contributions:** Investigation, methodology, formal analysis, data curation, writing—original draft preparation, writing—review and editing, S.S.; Conceptualization, supervision, project administration, visualization, writing—review and editing A.S. and P.K.N. investigation, methodology, supervision, writing—review and editing, N.K.T. All authors have read and agreed to the published version of the manuscript.

**Funding:** This research was supported by the Ministry of Food Processing Industries (MOFPI), New Delhi, India.

**Acknowledgments:** The authors are grateful to the Ministry of Food Processing Industries (MOFPI), New Delhi, India for granting the financial support; NIFTEM, Sonipat, Haryana, India for the institutional facility and National Collection of Dairy Cultures (NCDC), NDRI, India for providing the *Aspergillus niger*-224 culture.

**Conflicts of Interest:** The authors declare no conflict of interest.

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
