# Peer review of "A Comparative Study on the Debittering of Kinnow (Citrus reticulate L.) Peels: Microbial, Chemical, and Ultrasound-Assisted Microbial Treatment"

_fermentation, doi:10.3390/fermentation8080389_

Round 1

Reviewer 1 Report

Introduction

The authors have written a very good introduction to the topic of the article. They justified why they chose the subject of the study in the form of Kinnow mandarine peels, presented recent scientific achievements in the field of removing bitterness from citrus juices. The authors also justified the choice of three different methods used in the research presented in the article. Also, the presentation of the purpose and scope of the research work was very well and comprehensibly described.

Materials and Methods

2.2. Preparation of sample - Once the peel was pulverized, was it stored until analysis or did the authors process it further right away? Please, mention here in the text, how it was stored if such took place or provide information about immediate processing.

2.3.1. Chemical debittering of kinnow peels - Please give a brief general description of the procedure. Also state what your modification consisted of. Citing someone's methodology without providing details is not sufficient, and does not allow to reproduce it by others in the future.

2.3.2. Microbial debittering of kinnow peels by Aspergillus niger koji - Same comment as before. Provide a description of the methodology with an explanation of what the author's modifications consisted of.

2.3.3. Ultrasound-assisted microbial debittering of kinnow peels and 2.3.3.1 Experimental design for ultrasound-assisted microbial debittering – text from lines 137-145 should be swapped with text from lines 151-159. This is a better order of giving information to the reader and the titles of the subsections will correspond to their content.

2.4.1 Quantitative investigation of naringin by HPLC- Please give a full description of the preparation of the sample with the modifications made. Also provide the HPLC analysis parameters, mobile phase composition/proportions, etc. All this is needed for other researchers to reproduce the methodology.

2.4.2 Total phenol content (TPC), 2.4.3 Total flavonoid content (TFC), 2.4.4 Total antioxidant activity (TAA) - give a general description of the procedures for each spectrophotometric analysis with the wavelengths used. Most importantly, state the procedure for preparing polyphenol extracts from the obtained samples resulting from experiments to remove bitterness from mandarine peels. Did the authors make one extract and use it for all three analyses or did they make three different extractions, appropriate under each different spectrophotometric analysis?

3. Results and Discussion

The authors nowhere state the baseline naringin content in the processed mandarine peels, so it is impossible to fully determine which method of extracting this substance was the best. We don't know how much of this substance remained in the peels and whether any of the methods actually significantly removed the bitter naringin. The authors need to supplement the discussion of results with this information.

lines 252-254- this sentence needs citations and broader mention of these research. There is lack of any discussion of removing bitterness by chemical method.

line 278-284 - Did acting for such a long period of time with ultrasound on the mold-inoculated peels allow the mold to grow? Whether it was the effect of ultrasound alone and inoculation with mold did not matter due to low survival rate? How the temperature was regulated? - the ultrasonic treatment raises the temperature of the sample quickly.

3.4 Chemical composition of kinnow mandarin peels before and after debittering treatments- In my opinion, this subsection should be eliminated and the data on the initial content of naringin and testes substances/antioxidant activity moved to the beginning of the discussion chapter and incorporated wherever a method is evaluated. This will make it possible to assess how much a particular method has removed the naringin of its peels. 

line 483-484- I don't understand why the authors are quoting research results from another of their articles, which they should realize within the current article, because these results are necessary for the discussion. Quoting the article here is inappropriate.

4. Conclusions

The conclusions are based on the authors' research. However, there is a lack of information on which method was best for removing bitterness with justification. The authors focus too much in their conclusions on the RSM used for the ultrasound-assisted microbial debittering method. They should also elaborate in the conclusions on other threads of the experiments undertaken.

Author Response

All the reviewer's comments have been addressed and provided in the attached file.

Reviewer 2 Report

Dear Editor,

Thank you for the opportunity to review the manuscript entitled Comparative study on debittering of kinnow (Citrus reticulate L.) peels: Microbial, Chemical & Ultrasound-assisted microbial treatment. This study is very interesting with high possibility of application of the obtained results. The paper is well-structured, the literature is appropriate and the results are presented in a scientific manner. In my opinion, this manuscript should be accepted after minor revision. My only concerns refer to the part Materials and Methods:

Section 2.2. Indicate how the kinnow peels were cleaned?

Section 2.3.2. Line 130 erase extra dot after min

Section 2.3.2. Line 131 Indicate with changes were implemented in your study

Section 2.3.2. Line 132 Explain in more details how SSF and SMF was carried out

Section 2.3.3. Line 140 What is A. niger koji extract? How it is obtained?

Section 2.3.3. Line 145. My advice is to move Table 2 in Supplementary material

Section 2.3.3. How the values of independent variables were chosen (Table 1)? This missing information should be incorporated in the manuscript.

Author Response

(The authors gave the same response as above.)

Round 2

Reviewer 1 Report

I thank the authors for addressing all the comments and making many changes and improvements to the article. They all seem satisfactory.